# Infants’ Folate Markers and Postnatal Growth in the First 4 Months of Life in Relation to Breastmilk and Maternal Plasma Folate

**DOI:** 10.3390/nu15061495

**Published:** 2023-03-20

**Authors:** Rima Obeid, Ines Warnke, Igor Bendik, Barbara Troesch, Rotraut Schoop, Elodie Chenal, Berthold Koletzko

**Affiliations:** 1Department of Clinical Chemistry and Laboratory Medicine, Saarland University Hospital, D-66420 Homburg, Germany; 2DSM Nutritional Products Ltd., CH-4303 Kaiseraugst, Switzerland; 3Department of Paediatrics, The Ludwig Maximilian University of Munich (LMU), Dr. von Hauner Children’s Hospital and LUM University Hospitals, D-80337 Munich, Germany

**Keywords:** breastmilk, breastfeeding, folate catabolism, lactation, infant nutrition, (6S)-5-methyltetrahydrofolate

## Abstract

Background: Human milk is the sole source of folate in exclusively breastfed infants. We investigated whether human milk folate or maternal plasma folate are associated with infants’ folate status and postnatal growth in the first 4 months of life. Methods: Exclusively breastfed infants (n = 120) were recruited at age < 1 month (baseline). Blood samples were available at baseline and at the age of 4 months. Plasma and breastmilk samples were available from the mothers at 8 weeks postpartum. The concentrations of (6S)-5-methyltetrahydrofolate (5-MTHF) and different folate status markers were measured in samples of the infants and their mothers. The z-scores of weight, height, and head circumference of the infants were measured five times between baseline and 4 months. Results: Women with 5-MTHF concentrations in breastmilk <39.9 nmol/L (median) had higher plasma 5-MTHF compared to those with milk 5-MTHF concentrations >39.9 nmol/L (mean (SD) plasma 5-MTHF = 23.3 (16.5) vs. 16.6 (11.9) nmol/L; *p* = 0.015). At the age of 4 months, infants of women who were higher suppliers of 5-MTHF in breastmilk had higher plasma folate than those of low-supplier women (39.2 (16.1) vs. 37.4 (22.4) nmol/L; adjusted *p* = 0.049). The concentrations of breastmilk 5-MTHF and maternal plasma folate were not associated with infants’ longitudinal anthropometric measurements between baseline and 4 months. Conclusions: Higher 5-MTHF in breastmilk was associated with higher folate status in the infants and the depletion of folate in maternal circulation. No associations were seen between maternal or breastmilk folate and infants’ anthropometrics. Adaptive mechanisms might counteract the effect of low milk folate on infant development.

## 1. Introduction

Human milk is the optimal source of nutrition to support the development of the infant. The concentrations of folate remain constant in breastmilk during the first 3–5 months of lactation [1,2]. Folate concentrations decline in maternal circulation throughout lactation [3,4,5], and they increase in the circulation of the infant [6]. Folic acid supplementation has almost no effect on folate concentrations in breastmilk [2,4,7], suggesting that the secretion of folate into breastmilk already operates at a rather constant and nearly maximal capacity. However, the excretion of folate into breastmilk may deplete maternal folate, which could justify recommendations to supplement mothers with folate during lactation.

The adequate intake of folate for infants (0–6 months) was estimated at 65 µg/d based on concentrations of folate in breastmilk (i.e., mean 85 µg/L) [8,9] and assuming an average consumption of 0.78 L milk per day [10,11]. However, there are large methodological differences and between-population variations in milk folate concentrations [1,4,7,11]. Folate is mobilized from the mother to the infant to ensure the infant’s requirements [3,12,13]. There is no firm evidence that higher concentrations of folate in breastmilk as the sole source of folate would be associated with higher folate status in the infant [1,12]. As in adults, common polymorphisms in the 5,10-methylenetetrahydrofolate reductase (MTHFR) gene may influence folate markers and requirements in infants.

Intracellular folate is catabolized to para-aminobenzoylglutamate (pABG) that is excreted in urine after conversion to para-acetamidobenzoylglutamate (N-acetyl-pABG) [14,15,16,17]. The predominant form of folates in blood [18,19] and breastmilk [20,21] is 5-Methyltetrahydrofolate (5-MTHF). The 5-MTHF concentrations in breastmilk provide an estimate of folate intake in the infant and may thus show associations with the infant’s plasma concentrations of folate and pABG (a marker of tissue folate catabolism).

Folate is required for purine and pyrimidine synthesis in all growing cells. Higher folate intake during pregnancy is associated with a lower risk of preterm birth, low birth weight, and small-for-gestational-age births [22,23,24,25]. Therefore, it can be hypothesized that higher folate content in breastmilk or maternal plasma may be associated with the physical growth of breastfed infants.

We used data from 120 exclusively breastfed infants who formed a reference group in a previous randomized controlled trial [26]. The present explorative study investigated whether higher 5-MTHF concentrations in breastmilk or lower maternal plasma folate are associated with an infant’s higher folate markers at the age of 4 months. A secondary aim of the present study was to investigate whether higher breastmilk or maternal plasma folate are associated with infants’ z-scores of weight, height, and head circumference in the first 4 months of life.

## 2. Materials and Methods

### 2.1. Subjects

The present study included 120 exclusively breastfed infants and their mothers, who served as a reference group in a previous study [26]. The enrollment of the infants took place between June 2015 and April 2017 at the Department of Neonatology, Clinical Hospital Center “Dr. Dragiša Mišović-Dedinje”, Belgrade, Serbia. The inclusion criteria were: apparently healthy infants from singleton pregnancies who were younger than 1 month and were born between ≥37 and ≤42 weeks of gestation, had a birth weight between 2500 g and 4500 g, and parents/caregivers who intended to breastfeed for the duration of the study (4 months) and were able to speak the Serbian language. Infants were not eligible to participate if they had serious acquired or congenital diseases that may interfere with feeding or growth; feeding of more than 10% of energy (1 bottle/day) from sources other than breastmilk; abnormalities in hematological, hepatic, or renal markers; and the use of medication and vitamin supplements except for vitamin K or D or vaccination. The exclusion criteria for the mothers were diabetes mellitus, gestational diabetes, and adherence to a vegan diet that increases the risk of vitamin B12 deficiency in the infant.

The infants were recruited at a mean age of 20 and standard deviation (SD) of 3 days and were followed up until the age of 4 months (visit 4) without any intervention (Appendix A, study flow diagram). A medical examination was performed during the baseline visit. Infants’ weight, height, and head circumference were measured at baseline and repeated at a mean (SD) age of 29 (1) days, 56 (2) days, 85 (2) days, and 113 (3) days. Blood samples were collected from the infants at baseline and again at the age of 4 months.

Blood and breastmilk samples were collected from the mothers at 8 weeks postpartum. A questionnaire was used to assess the present use of maternal supplements containing folic acid.

The study was approved by the ethical committee at the University Hospital “*Dr Dragiša Mišović-Dedinje*” in Belgrade, Serbia. Written informed consent was obtained from parents or caregivers of the infants. The original study was registered at ClinicalTrials.gov (NCT02437721).

### 2.2. Blood and Breastmilk Analyses

In the mothers, fasting venous blood was collected into EDTAK+ tubes at 8 weeks postpartum. Venous blood samples were collected from the infants (at baseline and at the age of 4 months) into EDTAK+-containing tubes. Blood sampling was performed at least three hours after the last breast feeding. One aliquot of the EDTA-whole blood (50 µL) was immediately aspirated into an Eppendorf tube, and 450 μL of 1% ascorbic acid solution (*w*/*v* in bi-distilled H2O) were added to prepare blood hemolysate. The blood hemolysate samples were protected from light, mixed, and incubated for 30 min at room temperature and were then stored at −80 °C for measuring folate.

After an overnight fast, the women manually extracted approximately 5 mL of foremilk before the first feeding at home in the morning into a 10 mL container. The mothers transferred 3 portions of 1.5 mL each into Nunc^®^ CryoTubes^®^ (Thermo Fisher Scientific, Waltham, MA, USA) labeled with the subject ID as provided by the study center. The tubes were kept in household freezers at −20 °C for not more than 24 h before refrigerated transport to the study center, where they were stored at −80 °C until shipment to the laboratory.

All blood samples were centrifuged, and the EDTA-plasma was separated. The buffy coat from the infant’s blood was used for DNA isolation. All samples were directly frozen at −80 °C at the study center and then sent to the corresponding laboratories for further analyses. All lab procedures were performed according to standardized operating procedures. The lab staff members were blinded to participants’ data.

Plasma concentrations of total homocysteine (tHcy) were measured by a gas chromatography–mass spectrometry procedure at Bevital AS Laboratory (Bergen, Norway). The concentrations of folate forms and catabolites were measured in EDTA-plasma of the infants and the mothers using a liquid chromatography tandem MS/MS (LC-MS/MS) method at Bevital AS Laboratory [27]. The method depends on using corresponding ^13^C-labeled 5-MTHF, unmetabolized folic acid (UMFA), formyltetrahydrofolate, 4-alpha-hydroxy-5-methyltetrahydrofolate (hmTHF), pABG, and N-acetyl-pABG as internal standards as reported earlier [27]. The sum of the concentrations of 5-MTHF and hmTHF (an oxidation product of 5-MTHF) in plasma was considered as a surrogate marker of the total plasma folate. Only 6 women reported consumption of multivitamin supplements containing 500 µg folic acid taken once daily. UMFA was detectable in the plasma of 26 women (UMFA > 0.53 nmol/L). The concentrations of UMFA, formyltetrahydrofolate, and N-acetyl-pABG were very low and will not be addressed here.

The concentrations of 5-MTHF and UMFA in breastmilk were measured at DSM Nutritional Products using an LC-MS/MS method according to Page et al. [28]. In brief, folates were extracted with an acidic buffer followed by a multi-step tri-enzyme digestion using α-amylase, protease, and rat serum conjugase. Proteins were precipitated, and the folate extracts were purified using solid-phase extraction (SPE). Isotope-labeled internal standards and an external calibration curve were used for quantitative analysis of folate in milk. Folates were measured on a LC-MS/MS system in positive electrospray mode.

The folate concentrations of whole-blood hemolysate were measured at Bevital using a microbiological assay with a chloramphenicol-resistant strain of *Lactobacillus casei* [29]. The RBC-folate (nmol/L) concentrations were calculated using the following equation as established before [30]:RBC−folate=[(Whole blood hemolysate folate×10)− total plasma folate (1−Hct)]/Hct

The individual total plasma folate levels were approximated as the sum of plasma 5-MTHF and hmTHF levels, and hematocrit (Hct) results were used as decimal values (i.e., 0.40).

DNA was isolated from 110 infants with available buffy coat samples (Qiagen, Basel, Cat No. 51104), and the Applied Biosystems TaqMan SNP genotyping assay was used to determine the MTHFR C677T polymorphism (SNP rs1801133, Assay-ID: C_1202883_20) and the MTHFR A1289C polymorphism (SNP rs1801131, Assay-ID: C_850486_20).

### 2.3. Statistical Analyses

The descriptive data are shown as the mean (SD) for continuous variables and absolute (n) and relative frequencies (%) for categorical variables. A one-sample Kolmogorov–Smirnov Test with Lilliefors Significance Correction and Q-Q plots were used to study the distribution of the continuous variables. All folate markers were not normally distributed. Log_10_-transformation of the folate markers improved the distribution, as judged from the Q-Q plots. Hence, the log_10_-transformed values of folate markers and catabolites were used in all statistical analyses that assume normal distribution of the data.

Folate markers were compared between independent groups using ANOVA (one-way analysis of variance). ANOVA assumes random independent groups, normal distribution of the dependent variables, and homogeneity of the variance (i.e., *p* > 0.05 for Levene’s test of equality of variances within all groups). An ANCOVA test (ANOVA test with analysis of covariance) uses a regression analysis to adjust the difference between any 2 subgroups (i.e., high versus low breastmilk folate), with the subgroup as the explanatory variable and model-specific covariates.

ANCOVA was also used to compare the log_10_-transformed concentrations of folate parameters in the infants at each time point (i.e., at baseline and at 4 months) while adjusting for age and weight of the infant at the same visit. When the log_10_-transformed concentrations of folate parameters were compared at the age of 4 months, further adjustment for the log_10_-transformed concentrations of the same marker at baseline was performed. In addition, the log_10_-transformed concentrations of folate parameters at the age of 4 months were further adjusted for the infant’s genotype for the MTHFR C677T and A1289C polymorphisms. The adjustment for the MTHFR genotypes was due to the expected effect of the genotype on infant folate markers. For each ANCOVA model, we verified that the relationship between the dependent variable (i.e., the concentrations of a given folate marker) and the covariate (i.e., baseline levels of the same marker) was likely to be linear, and that there were no interactions between the breastmilk subgroups and the covariates.

Weight, length, and head circumference were standardized to z-scores, based on age, sex, and the WHO child growth standards, using the R-package version 0.3.1 [31]. Generalized estimating equations (GEE) were used to study the longitudinal associations between the concentrations of 5-MTHF (in breastmilk or in maternal plasma) or maternal plasma total folate with each of the outcomes (infant’s z-scores of weight, length, and head circumference between baseline and the age of 4 months). The GEE models with an unstructured correlation matrix were used. The dependent variable consisted of the longitudinal infant’s z-scores of weight, length, and head circumference (measured at baseline, visit 1, visit 2, visit 3, and visit 4). The visits constituted the within-subject variable. Separate models were run for each of the three anthropometric measures and each of the exposure variables. The GEE models were adjusted for the sex of the infant.

The statistical analyses were conducted using version 29 of IBM^®^ SPSS^®^ Statistics package (SPSS Inc., Chicago, IL, USA). *p*-values ≤ 0.05 were considered statistically significant, and values between 0.05 and 0.10 were considered to show a trend.

## 3. Results

The mean (SD) of birth weight in the infants was 3446 (383) g. The gestational age at birth was 39.6 (1.0) weeks. The age and the anthropometric measures of the infants at each of the study visits are shown in Table 1, and the study flow diagram is shown in Appendix A. Of the 110 infants with available DNA samples, 20% were homozygote for the MTHFR C677T polymorphism, and 9.1% were homozygote for the MTHFR A1298C polymorphism.

### 3.1. Concentrations of Folate Markers in Mothers and Their Breastfed Infants

The mean and (SD) of the concentrations of 5-MTHF in maternal plasma at 8 weeks postpartum were 20.0 (14.8) nmol/L, and 5-MTHF concentrations in breastmilk were 46.3 (23.3) nmol/L (Appendix A). In the infants, the plasma concentrations of 5-MTHF were 26.1 (15.5) nmol/L at baseline, and the levels increased to 34.3 (17.6) nmol/L at the age of 4 months (Appendix A). Total plasma folate (the sum of plasma 5-MTHF and hmTHF) was higher in the infants at baseline (35.1 (21.0) nmol/L) and at the age of 4 months (37.3 (19.5) nmol/L) compared to the concentrations of the mothers at 8 weeks postpartum (26.0 (17.2) nmol/L) (Appendix A). Concentrations of plasma pABG were higher in the infants compared to those in the mothers (8.5 (6.0) nmol/L in the infants at baseline and 14.8 (10.3) nmol/L at the age of 4 months versus 7.4 (5.5) nmol/L in the mother at 8 weeks postpartum) (Appendix A).

### 3.2. The Association between Breastmilk 5-MTHF and Folate Markers in the Mothers and Infants

We studied the associations between the concentrations of 5-MTHF in breastmilk (as a sole source of folate intake in the infant during the first 4 months of life) and the concentrations of folate markers in the mothers (at 8 weeks postpartum) and in the infants (at the age of 4 months). The concentrations of 5-MTHF in the breastmilk were dichotomized into low and high levels by the median of the whole group (39.9 nmol/L; min 13.2 nmol/L–max 126.1 nmol/L).

Higher concentrations of 5-MTHF in breastmilk (> vs. ≤39.9 nmol/L) were associated with lower maternal plasma concentrations of 5-MTHF (mean (SD) = 23.3 (16.5) vs. 16.6 (11.9) nmol/L, *p* = 0.015), plasma total folate (30.0 (19.4) vs. 21.9 (13.5) nmol/L, *p* = 0.007), and plasma pABG (8.5 (5.9) vs. 6.2 (4.8) nmol/L, *p* = 0.032) (Table 2).

The concentrations of 5-MTHF in breastmilk (> vs. ≤39.9 nmol/L) were not associated with infants’ plasma tHcy (adjusted *p* = 0.091) or RBC-folate (adjusted *p* = 0.103) at the age of 4 months (Table 2). In contrast, higher breastmilk 5-MTHF concentrations were associated with higher infants’ plasma total folate (5-MTHF plus hmTHF) (adjusted *p* = 0.049) (Table 2).

### 3.3. Maternal Plasma Folate, Breastmilk Folate, and Infant’s Folate Markers at Age of 4 Months

We studied whether maternal plasma total folate concentrations (i.e., defined as the sum of the concentrations of 5-MTHF and hmTHF; < vs. ≥20.3 nmol/L) were related to breastmilk folate or to infants’ folate markers. The concentrations of 5-MTHF in breastmilk tended to be lower in women with high plasma total folate compared to those with low plasma total folate (mean (SD) = 42.4 (23.3) nmol/L vs. 49.3 (23.2) nmol/L; *p* = 0.054) (Appendix A). Infants’ concentrations of plasma 5-MTHF, plasma folate catabolites, and RBC-folate at the age of 4 months did not differ according to maternal plasma folate (Appendix A).

### 3.4. Infants’ Longitudinal Anthropometrics in Relation to Breastmilk and Maternal Plasma Folate

The concentrations of 5-MTHF in breastmilk and maternal plasma were not associated with infants’ z-scores of weight, length, or head circumference between the baseline visit and the age of 4 months (Table 3).

## 4. Discussion

We investigated the association between folate markers in lactating women, breastmilk, and breastfed infants. We further studied whether breastmilk folate or maternal plasma folate were associated with the anthropometrical measurements of the infants between baseline and 4 months. Higher concentrations of 5-MTHF in breastmilk at week 8 postpartum were associated with depletion of folate from the mother’s circulation as shown by lower plasma 5-MTHF, plasma total folate, and plasma pABG. Higher concentrations of 5-MTHF in breastmilk were associated with generally higher folate in the infants at the age of 4 months. The concentrations of maternal plasma total folate were not associated with folate markers in the infants. Neither breastmilk 5-MTHF nor maternal plasma folate concentrations were associated with anthropometric measures of the infants between baseline and 4 months.

The vast majority of the lactating Serbian women in our study did not use supplemental folic acid, thus explaining the rather low maternal plasma folate concentrations compared to other populations [2,28]. A recent Chinese study reported that 5-MTHF concentrations in breastmilk were stable between 40 days and 400 days postpartum, and they did not differ between supplement users and nonusers throughout this period [32]. Stable folate concentrations in human milk throughout lactation have been reported by other investigators [1,2,33]. These results support that breastmilk folate can be considered as a surrogate marker of folate intake in the infants throughout the 4 months after birth in our study. We found considerable inter-individual differences in the concentrations of 5-MTHF in breastmilk. The mean value of 5-MTHF in milk (46.3 nmol/L equivalent to approximately 20 µg folate/L) was markedly lower than those in other populations (64 μg/L to 85 µg/L) [7,8,9,32,34]. However, for unknown reasons, the variations in concentrations of 5-MTHF in breastmilk did not translate into similar variations in the folate status of the infants.

Infants of high folate-supplier mothers (breastmilk 5-MTHF concentration >39.9 nmol/L; range: 40.2 nmol/L to 126.1 nmol/L or 18 µg/L to 58 µg/L) had a mean body weight of 6.6 kg at the age of 4 months, and they achieved between 14 µg and 45 µg folate intake per day, assuming that the infants would consume 0.78 L milk per day. Infants receiving milk from women who were low folate suppliers (breastmilk 5-MTHF ≤ 39.9 nmol/L or ≤18 µg/L) would achieve a daily folate intake below 14 µg/day. Tamura et al. reported a weak association between breastmilk 5-MTHF and infant folate status [7]. This suggests that in addition to the amount of 5-MTHF in breastmilk, the absorption and bioavailability of folate from breastmilk may determine the folate status of the infants. For example, there could be variations in the volume of breastmilk consumed by the infant or in the levels of folate-binding proteins in breastmilk. Folate-binding proteins may regulate intestinal folate absorption and bioavailability in the infant [35,36,37] and may be subject to influences from other milk components such as macronutrients [38]. In addition, the MTHFR C677T and A1298C polymorphisms could determine folate status under certain folate intake levels [39]. In our study, the number of infants who were homozygote for the MTHFR C677T and A1298C variants was not sufficient for detailed investigations of the effect of the genotype on folate markers in the infants.

In contrast to our results of an inverse association between breastmilk and maternal plasma 5-MTHF concentrations (Table 2), studies in women with high folate intake (from supplements and fortified foods) found no such associations [2,4,7]. This is in line with earlier studies showing that the excretion of recently taken folate into breastmilk is favored, even when the mother is deficient [40]. Therefore, folate supplementation during lactation can ensure an adequate supply of folate to prevent the depletion of maternal folate and provide sufficient folate intake to their breastfed infants. The results of the present study are likely to be generalizable to European populations in which there is no fortification of staple foods with folic acid in place, and women do not regularly use folate-containing multivitamins beyond the first trimester.

We found no associations between concentrations of 5-MTHF in breastmilk or in maternal plasma at 8 weeks postpartum and the changes in infants’ anthropometric measures from baseline to the age of 4 weeks. There is mixed evidence from the literature on the association between maternal folate and pre- or postnatal growth. Higher maternal folate concentrations (especially RBC-folate) during pregnancy have been reported to be associated with larger head circumference of the neonates at birth [41,42]. The positive association between maternal RBC-folate, folic acid supplement use, or dietary folate intake and infants’ birth weight seems to be consistent across studies [43,44]. Our study was not originally designed to detect variations in infants’ growth according to breastmilk folate. To test this hypothesis, we might need a larger samples size or a higher percentage of women taking multivitamin supplements with folate. It is possible that breastmilk folate was not low enough to affect infants’ growth. In addition, low folate intake during pregnancy may make the infants more resistant to low folate intake after birth by mechanisms related to foetal programming.

The present study has a few limitations. First, this is an exploratory analysis of data collected for a suitability and safety study. Second, the concentrations of RBC-folate and plasma tHcy and the MTHFR genotype were not available from the mothers. These markers may explain variations in 5-MTHF in breastmilk. Third, measurement of breastmilk folate is subject to several sources of variations. Breastmilk folate may increase up to 3 months [12]. However, previous studies found rather stable concentrations of folate in breastmilk in the first 4 months postpartum [2,32], suggesting that the milk content of folate at visit 2 (8 weeks postpartum) may be representative of the average intake of the infant in the first 3 months. In addition, we measured 5-MTHF in foremilk samples collected in the morning before the start of the lactation session, but we do not have data to show whether breastmilk folate is stable through one lactation session. This needs to be investigated in future studies.

## 5. Conclusions

We found that higher concentrations of 5-MTHF in breastmilk were associated with lower 5-MTHF in the plasma of the mother, suggesting that lactation depletes maternal folate. The concentrations of folate status markers of breastfed infants at the age of 4 months were generally higher in infants of mothers supplying high levels of 5-MTHF in breastmilk compared to infants of women who were low suppliers of 5-MTHF in breastmilk. Changes in the anthropometric measures of the infants between baseline and 4 months were not related to breastmilk 5-MTHF or maternal plasma folate. Measurement of folate-binding proteins in human milk may clarify mechanisms that could enhance folate absorption and bioavailability in the case of low maternal folate status.

## Figures and Tables

**Table 1 nutrients-15-01495-t001:** Characteristics and anthropometric measures ^1^ of the breastfed infants.

	Mean (SD)	Mean (SD) of Z-Scores
Birth weight, g, n = 120	3446 (383)	
Gestational age at birth, weeks, n = 120	39.6 (1.0)	
Male sex, n (%), n = 120	58 (48.3%)	
Baseline visit, n = 120		
Age, days	20 (3)	
Weight, g	3905 (425)	0.004 (0.702)
Length, cm	54.0 (1.9)	0.681 (0.963)
Head circumference, cm	36.2 (1.1)	0.116 (0.862)
Visit 1, n = 115		
Age, days	29 (1)	
Weight, g	4299 (441)	0.045 (0.705)
Length, cm	55.2 (1.9)	0.628 (0.924)
Head circumference, cm	37.0 (1.1)	0.191 (0.868)
Visit 2, n = 114		
Age, days	56 (2)	
Weight, g	5267 (530)	0.073 (0.702)
Length, cm	58.4 (2.0)	0.600 (0.906)
Head circumference, cm	38.8 (1.2)	0.309 (0.892)
Visit 3, n = 112		
Age, days	80 (2)	
Weight, g	6034 (625)	0.077 (0.761)
Length, cm	61.7 (2.1)	0.803 (0.966)
Head circumference, cm	40.2 (1.2)	0.364 (0.856)
Visit 4, n = 112		
Age, days	113 (3)	
Weight, g	6647 (681)	0.095 (0.794)
Length, cm	64.2 (2.1)	0.884 (0.932)
Head circumference, cm	41.4 (1.2)	0.488 (0.883)
MTHFR C677T genotype ^2^	n = 110 (100%)	
CC, n	40	
CT, n	48	
TT, n	22	
MTHFR A1298C genotype ^2^	n = 110 (100%)	
AA, n	70	
AC, n	30	
CC, n	10	

^1^ Weight, length, and head circumference were standardized to z-scores, based on age, sex, and the WHO child growth standards. ^2^ The MTHFR C677T and MTHFR A1298C polymorphisms were determined in 110 infants with available blood samples for DNA isolation. MTHFR, 5, 10-methylenetetrahydrofolate reductase.

**Table 2 nutrients-15-01495-t002:** Concentrations of folate status markers and catabolites in lactating women and their infants according to breastmilk concentrations of 5-MTHF dichotomized by the median.

**Maternal Folate Markers (8 Weeks Postpartum)**	**Low Suppliers of 5-MTHF in Breastmilk** **Mean (SD) = 29.1 (7.6)** **Range 13.2–39.6 nmol/L ^1^**		**High Suppliers of 5-MTHF in Breastmilk** **Mean (SD) = 62.9 (21.5)** **Range 40.2–126.1 nmol/L ^1^**		***p* ^2^ (Low vs. High Breastmilk Folate)**		
Plasma 5-MTHF, nmol/L	23.3 (16.5)		16.6 (11.9)		0.015		
Plasma pABG, nmol/L	8.5 (5.9)		6.2 (4.8)		0.032		
Plasma N-acetyl-pABG, nmol/L	0.7 (0.3)		0.7 (0.3)		0.684		
pABG + N-acetyl-pABG, nmol/L	9.2 (6.0)		7.0 (5.0)		0.034		
Plasma hmTHF, nmol/L	6.6 (3.5)		5.3 (2.4)		0.015		
Sum of 5-MTHF and hmTHF in plasma, nmol/L	30.0 (19.4)		21.9 (13.5)		0.007		
**Infant’s Folate Markers**	**Baseline**	**V4**	**Baseline**	**V4**	**Baseline *p* ^3^**	**Visit 4** ***p* ^4^**	**Visit 4** ***p* ^5^**
Plasma 5-MTHF, nmol/L	26.1 (15.4)	33.5 (19.9)	25.9 (16.2)	34.8 (14.9)	0.983	0.175	0.065
Plasma pABG, nmol/L	8.7 (5.4)	13.4 (8.9)	7.9 (6.5)	16.1 (11.5)	0.141	0.025	0.114
Plasma N-acetyl-pABG, nmol/L	1.1 (0.3)	0.6 (0.1)	1.1 (0.4)	0.7 (0.3)	0.371	0.165	0.076
Plasma pABG + N-acetyl-pABG, nmol/L	9.8 (5.3)	14.0 (8.9)	9.0 (6.5)	16.8 (11.5)	0.180	0.026	0.110
pABG + N-acetyl-pABG nmol/L/kg body weight	2.5 (1.4)	2.3 (1.4)	2.3 (1.8)	2.8 (1.9)	0.150	0.027	0.113
Plasma hmTHF, nmol/L	9.1 (6.4)	3.9 (3.4)	9.2 (5.9)	4.3 (2.4)	0.819	0.026	0.025
Sum of 5-MTHF and hmTHF in plasma, nmol/L	35.2 (21.0)	37.4 (22.4)	35.1 (21.4)	39.2 (16.1)	0.931	0.132	0.049
Plasma tHcy, µmol/L	6.7 (1.9)	8.7 (2.8)	7.0 (2.3)	8.6 (2.9)	0.394	0.088	0.091
RBC-folate, nmol/L	1365 (486)	1292 (673)	1427 (631)	1391 (528)	0.985	0.095	0.103

Data are shown as mean (SD). Mean (SD) of infant’s age was <1 month at baseline and 4 months at visit 4. ^1^ Breastmilk concentrations of 5-MTHF were dichotomized by the median value in the 113 mothers (39.9 nmol/L: min. 13.2 nmol/L and max. 126.1 nmol/L). ^2^ *p* values are from ANOVA test applied on the log_10_-transformed data. ^3^ Log_10_-transformed concentrations at baseline visit were compared between strata of breastmilk folate by using ANCOVA with age and weight of the infant at baseline as covariates. ^4^ Log_10_-transformed concentrations at 4 months were compared using ANCOVA test with age and weight of the infant at age of 4 months and the baseline concentrations of the same marker as covariates. ^5^ ANCOVA test for folate concentrations at age of 4 months was further adjusted for the genotype of the MTHFR C677T and A1298C polymorphisms in the infants.

**Table 3 nutrients-15-01495-t003:** The association between breastmilk 5-MTHF or maternal plasma folate at 8 weeks postpartum and the longitudinal changes in z-scores of infant’s weight, length, and head circumference between the age < 1 month and 4 months.

Exposure Variable	Z-Scores of Weight ^2^	Z-Scores of Length ^2^	Z-Scores of Head Circumference ^2^
Log-breastmilk 5-MTHF			
Unadjusted β (95% CI), n = 566 observations	0.096 (−0.535, 0.728)	0.160 (−0.571, 0.891)	0.564 (−0.165, 1.292)
Adjusted β (95% CI) ^1^	0.097 (−0.522, 0.717)	0.160 (−0.574, 0.894)	0.573 (−0.159, 1.305)
Log−5-MTHF in plasma of the mother			
Unadjusted β (95% CI), n = 566 observations	−0.096 (−0.508, 0.316)	−0.020 (−0.443, 0.397)	−0.259 (−0.754, 0.237)
Adjusted β (95% CI) ^1^	−0.099 (−0.502, 0.304)	−0.023 (−0.444, 0.397)	−0.258 (−0.754, 0.237)
Log-sum of 5-MTHF and hmTHF in plasma of the mother			
Unadjusted β (95% CI), n = 561 observations	−0.170 (−0.645, 0.304)	−0.053 (−0.550, 0.445)	−0.403 (−0.992, 0.187)
Adjusted β (95% CI) ^1^	−0.171 (−0.630, 0.288)	−0.053 (−0.551, 0.445)	−0.404 (−0.993, 0.185)

Generalized estimating equation (GEE) models were applied using one infant anthropometric measure (as z-score) for each model as an outcome variable (5 measurements per infant). The regression coefficient β and (95% confidence intervals; 95% CI) show the change in z-scores of infants’ weight, height, and head circumference for each 1-unit change in the log-transformed exposure variables. ^1^ Adjusted for infant sex (boys, girls). ^2^ Infants anthropometrics were measured at the study center at baseline visit (age < 1 month), visit 1, visit 2, visit 3, and visit 4 (age 4 months); weight was measured in g, and length and head circumference in cm.

## Data Availability

The authors declare that they will share data in aggregated form for research purposes upon request, under conditions respecting the EU General Data Protection Regulation and the protection of personal rights of study subjects.

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
