# Peer review of "Infants’ Folate Markers and Postnatal Growth in the First 4 Months of Life in Relation to Breastmilk and Maternal Plasma Folate"

_nutrients, 2023, doi:10.3390/nu15061495_

Round 1

Author Response

Point by point reply to the comments of the reviewers

Infant’s folate markers and postnatal growth in the first 4 months of life in relation to maternal plasma- and breastmilk and maternal plasma folate. Obeid et al.,

Reply to Reviewer 1

This study focuses on possible correlations between infant’s folate status and postnatal growth to maternal plasma and breastmilk folate in exclusively breastfed infants. Authors found that the concentrations of folate markers of breastfed infants aged 4 months were higher if the mothers had higher breastmilk folate, but were not related to maternal plasma folate, they also found that changes of the anthropometric measures of the infants between age 20 days and 4 months were not related to breastmilk 5-MTHF or maternal plasma folate. The topic is interesting but I have some observations:

Reviewer Comment

Authors Reply

Line 3 I suggest to add “of life” after the word “months”

Done

and please eliminate “-” after the word “plasma”.

Done

Abstract: the aim of the study is missing and please improve the Background.

Human milk is the sole source of folate in exclusively breastfed infants. The study aimed to investigate whether maternal plasma folate and human milk folate could affect infant’s folate status and postnatal growth in the first 4 months of life. 

Line 18-19 please move “[mean (SD) = 23.3 18 (16.5) vs. 16.6 (11.9) nmol/L; p = 0.015]” at the end of the sentence.

Done

Line 90 please move (standard deviation, SD) before number 3.

Done

Line 91 Figure S1, Study Flow Diagram is missing. Line 268 Table S1 is missing.

Now attached

Line 94 it is no necessary to repeat (SD).

Removed

Analysis of MTHFR gene polymorphisms was reported but the obtained data were not discussed please add them in results and discussion.

We revised to briefly address this idea. The study is explorative and the number of infants who were carriers of the homozygote genotypes is low and does not allow in depth analysis of the effect of this genetic factor.  We briefly referred to this point in the discussion.

A lot of references are outdated, please replace them with more recent ones

We removed 2 references. The literature on this topic is relatively old. The studies that were used to set the dietary recommendations for folate in infants should be cited. To the best of our knowledge there are no comparable new studies.

Reviewer 2 Report

The paper titled Infant’s folate markers and postnatal growth in the first 4 2 months in relation to maternal plasma- and breastmilk folate  studied the association between 14 the concentrations of (6S)-5-methyltetrahydrofolate (5-MTHF) in breastmilk and maternal plasma 15 folate at 8 weeks postpartum and folate markers and the z-scores of weight, height and head cir- 16 cumference of the infants between age < 1 month and 4 months. Paper is very intersting but also confusing, so need some changes and improvements. The main problem is genetics ,authors have same genotyping results but those results are not discused in discussion and alsoin introduction authors dont mentioned genetics at all. Why you done  this genetics onlyin infants? Genotyping mothers is also very important for better explain this topic, way some women were low suppliers of 5-MTHF in breastmilk?

Introduction section -

line 59 do 72 - is sutible for section materials and methods in that line described hypotesis and aim of the study.

Results:

Think about to reduce text or  tables

Discussion  need to be improved with genetics if you think that those results are obligate for the study.

Author Response

Point by point reply to the comments of the reviewers

Infant’s folate markers and postnatal growth in the first 4 months of life in relation to maternal plasma- and breastmilk and maternal plasma folate. Obeid et al.,

Reply to Reviewer 2

The paper titled Infant’s folate markers and postnatal growth in the first 4 2 months in relation to maternal plasma- and breastmilk folate studied the association between 14 the concentrations of (6S)-5-methyltetrahydrofolate (5-MTHF) in breastmilk and maternal plasma 15 folate at 8 weeks postpartum and folate markers and the z-scores of weight, height and head cir- 16 cumference of the infants between age < 1 month and 4 months.

Paper is very interesting but also confusing, so need some changes and improvements.

Reviewer Comment

Authors Reply

The main problem is genetics ,authors have same genotyping results but those results are not discused in discussion and alsoin introduction authors dont mentioned genetics at all. Why you done  this genetics onlyin infants?

We fully agree that we need some information in the introduction and discussion on the genetic polymorphisms tested in this cohort. We have done that in the present revised version.

However, we the article should not be focused on genetic polymorphism, because of the low number of participants and because we have recently published a larger study on this topic with 290 infants and found associations with child folate markers (the study is currently in press- Am J Clinical Nutrition). So there are sufficient reasons for us to think that we should adjust for the genetic variants when we study the association between folate intake from human milk and status markers on the infant. 

Genotyping mothers is also very important for better explain this topic, way some women were low suppliers of 5-MTHF in breastmilk?

We fully agree that the MTHFR genotype of the mother is important. However, this a completely different question since we are primarily looking at human milk folate as a source of folate intake in the infant and infant’s folate markers.

The original study was planned as a safety study for infant’s formula. So we did not plan to do genetic investigations on the mothers.

Introduction section - line 59 do 72 - is sutible for section materials and methods in that line described hypotesis and aim of the study.

We modified the introduction as suggested

Results:

Think about to reduce text or  tables

We revised the article and reduced as much as possible of the text. Table 2 in the previous version is now removed to the supplemental Table S1 (one table is now less in the text of the article). All other tables are necessary for a minimal level of transparency.

Discussion  need to be improved with genetics if you think that those results are obligate for the study.

We briefly discussed the genetic aspects of the MTHFR polymorphisms.   

Round 2

Reviewer 2 Report

NA